# Improvement of Enzymatic Saccharification and Ethanol Production from Rice Straw Using Recycled Ionic Liquid: The Effect of Anti-Solvent Mixture

**DOI:** 10.3390/bioengineering9030115

**Published:** 2022-03-11

**Authors:** Santi Chuetor, Elizabeth Jayex Panakkal, Thanagorn Ruensodsai, Kraipat Cheenkachorn, Suchata Kirdponpattara, Yu-Shen Cheng, Malinee Sriariyanun

**Affiliations:** 1Department of Chemical Engineering, Faculty of Engineering, King Mongkut’s University of Technology North Bangkok, Bangkok 10800, Thailand; santi.c@eng.kmutnb.ac.th (S.C.); kraipat.c@eng.kmutnb.ac.th (K.C.); suchata.k@eng.kmutnb.ac.th (S.K.); 2Department of Chemical and Process Engineering, The Sirindhorn International Thai-German Graduate School of Engineering, King Mongkut’s University of Technology North Bangkok, Bangkok 10800,Thailand; elizabeth.jayex@gmail.com (E.J.P.); oventhanagorn@gmail.com (T.R.); 3Department of Chemical and Materials Engineering, National Yunlin University of Science and Technology, Yunlin 640, Taiwan; yscheng@gemail.yuntech.edu.tw

**Keywords:** anti-solvent, enzymatic saccharification, EMIM-Ac, ethanol, ionic liquid, lignocellulosic biomass, recycling

## Abstract

One of the major concerns for utilizing ionic liquid on an industrial scale is the cost involved in the production. Despite its proven pretreatment efficiency, expenses involved in its usage hinder its utilization. A better way to tackle this limitation could be overcome by studying the recyclability of ionic liquid. The current study has applied the Box–Behnken design (BBD) to optimize the pretreatment condition of rice straw through the usage of 1-ethyl-3-methylimidazolium acetate (EMIM-Ac) as an ionic liquid. The model predicted the operation condition with 5% solid loading at 128.4 °C for 71.83 min as an optimum pretreatment condition. Under the optimized pretreatment condition, the necessity of the best anti-solvent was evaluated among water, acetone methanol, and their combinations. The study revealed that pure methanol is the suitable choice of anti-solvent, enhancing the highest sugar yield. Recyclability of EMIM-Ac coupled with anti-solvent was conducted up to five recycles following the predicted pretreatment condition. Fermentation studies evaluated the efficacy of recycled EMIM-Ac for ethanol production with 89% more ethanol production than the untreated rice straw even after five recycles. This study demonstrates the potential of recycled ionic liquid in ethanol production, thereby reducing the production cost at the industrial level.

## 1. Introduction

The growing urbanization and the tremendous growth of industries have led to a rise in energy requirements. However, the production of energy from non-renewable origins as the main source is debated worldwide regarding its impact on environmental sustainability. Therefore, renewable energy in various forms has gained popularity and importance in policies of governmental and private sectors. Lignocellulosic biomass, being more abundant in nature, is considered the major biological feedstock for energy production. Globally, it is estimated that nearly 180 billion tons of lignocellulosic biomass are produced annually, having the potential to be converted into various high value-added goods, including biofuels, biochemicals, and biomaterials via the biorefinery process [1]. Lignocellulosic [2,3] material comprises cellulose, hemicellulose, and lignin as biopolymeric constituents that can be broken down into reactive biomolecules, which are transformed into valuable products and fuels [4].

However, the utilization of lignocellulose biomass as raw material has several limitations, including its physical and chemical barriers, the crystallinity of cellulose, and complex and recalcitrant structures [5,6]. An inevitable process is a pretreatment that basically enhances the lignocellulosic valorization. The efficient pretreatment technology must render the lignocellulosic material more reactive to subsequent processes [7]. Moreover, the hard barrier of using lignocellulosic as feedstocks are the chemical and physical properties that hinder the hydrolysis reaction.

To overcome these limitations, the selection of efficient pretreatment should be considered as a crucial factor for lignocellulosic valorization. The deficient pretreatment technique results in lower conversion yields that consequently determine economic infeasibility on a high production scale. Aforementioned argumentation, the selection of lignocellulosic biomass pretreatment is, therefore, necessary to reduce the recalcitrance of lignocellulosic biomass structure and to eliminate some inhibitors including; 5-hydroxyethyl furfural, furfural, and acetic acid that are produced by sugar degradation during pretreatment. An efficient pretreatment should provide reactive material for further processes, including enzymatic accessibility and fermentation. To reach a high conversion yield, the presence of lignin components in the lignocellulosic structure is unnecessary due to its specific characteristic for hindering the enzymatic accessibility [8].

Pretreatment processes are generally grouped into four categories, namely chemical, physical, biological, and physicochemical pretreatment [9,10]. Each pretreatment technique affects different properties, including chemical and physical properties. As previously mentioned, the effects of these pretreatments are demonstrated in various lignocellulosic biomasses, which could be helpful for their application in the biorefining process [11]. Among the various pretreatment process, the most widely used process is chemical pretreatment because of its ability to alter the biopolymeric conformation of the biomass and its uncomplicated application compared to other pretreatment methods [11]. Chemical pretreatment helps in removing the chemical linkages associated with three biomolecules together that it facilitates further processes [12]. Various existing chemical pretreatment methods, including alkaline, acid, deep eutectic solvent extraction, and ionic liquid, were previously investigated to enhance sugar production and bioethanol production [13]. However, their application on an industrial scale is limited due to the investment cost. Pretreatment and hydrolysis processes are considered a high cost and time investment from an industrial point of view [14,15]. Hence, it is preferable to select a potential pretreatment method, which can reduce the total cost for its industrial application. Among the various types of chemical pretreatment techniques, a widely advanced technique is ionic liquid pretreatment, which has the potential to be recycled and reused.

Ionic liquids (ILs) were recently recommended for lignocellulosic pretreatment because of their properties such as high heat resistance, high chemical stability, low liquefaction point, noninflammable, high polarity, and less risky process [10,16]. The ILs mainly emphasize lignin removal and cellulose structural swelling [17]. They also have other applications in various types of lignocellulosic materials for enhancing conversion [17]. One important property of ILs that makes them attractive for pretreatment purposes is their recyclability [18,19]. The recycling capacity of the IL can help in reducing the cost of the pretreatment. The low volatility of IL makes it feasible for recycling studies [20]. The integration of IL to the biorefining process was also previously demonstrated to be compatible with IL-tolerant cellulases [21,22]. Additionally, cellulose present in the biomass pretreated with IL can be hydrolyzed easily when various anti-solvents such as water, acetone, ethanol, methanol, and dichloromethane are used in the washing process [23]. In fact, the efficacy of ionic liquid pretreatment does not only depend on the type of ionic liquid but also on the anti-solvent and pretreatment condition [24]. After the pretreatment process, anti-solvent can be removed easily through the process of evaporation, and the ionic liquid is retrieved for the next round of pretreatment. A previous study on pretreatment of cotton with EMIM-Ac and its recyclability showed that the ionic liquid maintained its efficiency even after five recycles in comparison with its initial usage [25]. Another study on the pretreatment of spruce and oak dust with EMIM-Ac revealed an enhancement in the sugar yield post pretreatment and the ionic liquid could be reused up to seven times [20]. However, another recycling study with ionic liquid, 1-ethyl-3-methylimidazolium acetate and 1-ethyl-3-methylimidazolium acetate/ethanolamine, showed that reused IL was less efficient after 5–7th recycling [26]. The study also revealed that the lignin and moisture content in the recycled ionic liquid could influence pretreatment. The study carried out on the pretreatment of wood meals with 1-ethyl-3-methylimidazolium acetate and its recyclability showed that the pretreatment with recycled IL was effective until three recycles even without the removal of the accumulated lignin after each recycle [27]. Even though researchers studied the recyclability of IL, there are few reports of the recyclability effect on bioethanol production from recycled IL.

Hence, this study focused on evaluating the performance of the IL, 1-ethyl-3-methylimidazolium acetate (EMIM-Ac) pretreatment for bioethanol production with recycled ionic liquid-solvent mixture under optimal pretreatment conditions. Optimization studies were conducted with a mathematical model (BBD) that could predict the optimum pretreatment condition based on the interaction between variables and factors. The study also focused on recycling EMIM-Ac and its effect on rice straw pretreatment to improve the reducing sugar and ethanol yield.

## 2. Materials and Methods

### 2.1. Biomass Preparation

Rice straw was procured from a local paddy field in the central part of Thailand. The moisture content from the collected biomass was removed by drying it in a hot-air oven (WOF-50, Daihan Scientific, Gangwon-do, Korea) at 80 °C until a constant weight was obtained. Then, the dried rice straw sample was reduced in size using a household blender and sieved through a 20-mesh-sized aluminum sieve to obtain a uniform particle size. The biomass composition of the rice straw samples was analyzed by following the Van Soest protocol [28].

The 1-ethyl-3-methylimidazolium acetate (EMIM-Ac) and commercial cellulase enzyme, CelluClast 1.5 L, produced by *Trichoderma reesei* used in this study was bought from Sigma-Aldrich (St. Louis, MO, USA). The enzyme β-glucosidase from *Aspergillus niger* was obtained from Megazyme (Wicklow, Ireland). The 3,5-Dinitrosalicyclic acid used in reducing sugar determination was purchased from Alfa Aesar (Heysham, UK). The other solvents used in this study were obtained from RCI Labscan (Bangkok, Thailand).

### 2.2. Optimization of Rice Straw Pretreatment for Reducing Sugar Production

A mathematical model from response surface methodology (RSM), namely, the Box–Behnken design (BBD), was employed for the optimization of pretreatment conditions using EMIM-Ac. The model considered three pretreatment factors, namely, solid loading ratio (X_1_: 5–15 wt%), pretreatment temperature (X_2_: 100–140 °C), and pretreatment time (X_3_: 30–60 min). Each of these pretreatment factors was tested at 3 levels, high (+1), mid (0), and low (−1) (Table 1). The model predicted a total of 17 runs, from among which the optimal pretreatment condition was selected based on the highest sugar yield (Y). Design-Expert software version 7.0.0 (STAT-EASE Inc., Minneapolis, MN, USA) was used to analyze the optimization result [29]. The effects of each pretreatment factor and interacting effects between two factors on the highest sugar yield were analyzed by ANOVA with the significance level of *p*-value less than 0.05.

### 2.3. EMIM-Ac Pretreatment Procedure

Pretreatment was performed following the suggested pretreatment conditions from BBD. Rice straw pretreatment was conducted by mixing 1 g of dried rice straw with 19, 9, and 5.67 g of EMIM-Ac to obtain a solid loading ratio at 5, 10, and 15 wt%, respectively, in a screw-capped tube. Then, every reaction was proceeded to a targeted pretreatment temperature (100–140 °C) in a controlled-temperature hot air oven (±2.0 °C, Model: WOF-50, Daihan Scientific, Gangwon-do, Korea) with the retention time of the targeted pretreatment time (30–90 min). After the pretreatment, the sample was taken out from the hot air oven and quickly cooled down to room temperature by placing the tube in a room-temperature water bath (Model: WB-22, Daihan Scientific, Gangwon-do, Korea) for 5 min to stop the progress of pretreatment effect. The solids in the pretreated biomass were regenerated by the addition of water as an anti-solvent in a 1:1 (w/w) ratio. Then, the pretreated biomass was separated into a solid and liquid fraction by centrifugation (Combi 514R, Hanil Scientific Inc., Gimpo, Korea) at 8000× *g* for 10 min. The separated solids after pretreatment were washed thoroughly with 50 mL of deionized water three times to remove any EMIM-Ac residues. Then, the pretreated solids were oven-dried in a controlled-temperature hot air oven (±2.0 °C, Model: WOF-50, Daihan Scientific, Gangwon-do, Korea) at 80 °C until moisture was removed completely and a constant weight was achieved. The solid biomass void of moisture was subsequently hydrolyzed enzymatically, and the amounts of reducing sugars released by hydrolysis of biomass were analyzed using the dinitrosalicylic method (DNS) [30]. The liquid fractions containing EMIM-Ac were collected after centrifugation and were proceeded to the recycling process for the next round of pretreatment (Figure 1).

### 2.4. Recyclability of EMIM-Ac with Different Anti-Solvents

The EMIM-Ac recyclability studies were conducted by using 3 kinds of anti-solvents, comprising deionized water, acetone, and methanol. After the pretreatment of the rice straw sample, each type of anti-solvent was added to the mixture with the ratio of 1:1 (*w*/*w*). Solid and liquid fractionation was carried out by centrifugation as described in Section 2.3 (Figure 1). The liquid fraction containing EMIM-Ac was then filled in a rotary evaporator (Model: RV 10DS93, IKA, Tokyo, Japan) to recover EMIM-Ac and remove anti-solvents by setting the temperature of the water bath at 100, 56, and 65 °C for water, acetone, and methanol, respectively. The weight of the remaining liquid in the rotary evaporator was measured every 10 min, and the recycled EMIM-Ac was collected when the weight was consistent. The recycled EMIM-Ac was re-used further for the pretreating process of the new rice straw sample at the optimum pretreatment condition obtained from the RSM experiment for 5 rounds of recycling.

### 2.5. Enzymatic Hydrolysis and Measurement of a Sugar Yield

The efficacy of the EMIM-Ac pretreatment was estimated by enzymatic hydrolysis of the pretreated sample and measurement of reducing sugar concentration released from the hydrolyzed biomass based on the protocol published in our previous works [31,32]. An enzymatic hydrolysis reaction was set up in a screw-capped tube with 0.1 g biomass (2.5% *w*/*v*) in 4 mL of 50 mM citrate buffer (pH 4.7). A mixture of commercially available cellulase enzymes, CelluClast 1.5 L, was added into the mixture at the concentration of 20 FPU/g-biomass. To avoid any microbial contamination during hydrolysis reaction, 40 µL of 2 M sodium azide (Ajax Finechem, Brooklyn, MA, USA) was supplemented to the reaction. The enzymatic reaction was incubated at 45 °C for 72 h with 200 rpm agitation in a shaking incubator (JSSI-100C, JS Research Inc., Gongju, Republic of Korea). Any progress in the hydrolysis reaction was stopped by incubating at 100 °C for 5 min in a water bath (Model: WB-22, Daihan Scientific, Gangwon-do, Korea). The hydrolyzed sample was separated into a solid and liquid fraction by centrifugation at 8000× *g* for 10 min. The amount of reducing sugars in liquid hydrolysates was quantified using the 3,5-dinitrosalisylic acid (DNS) method [30]. The hydrolyzed solid fraction was collected and oven-dried at 80 °C until moisture was completely removed and constant weight was recorded. All the experiments were performed in triplicates.

### 2.6. Fermentation and Analysis of Ethanol Yield

The impact of IL pretreatment at the conversion of rice straw to bioethanol was observed by a fermentation experiment using *Saccharomyces cerevisiae* with the procedure used in our previous studies [10,33]. Fermentation was carried out using a liquid hydrolysate fraction obtained from enzymatic hydrolysis without the addition of sodium azide (Section 2.4), eliminating the supplementation of sodium azide. Yeast inoculum (1 mL at the concentration of about 10^8^ cells/mL) was inoculated into 19 mL biomass liquid hydrolysate supplemented with glucose (1%, *w*/*v*) and yeast extract (1%, *w*/*v*). The fermentation mixture containing the yeast culture was incubated in a rotary shaker (Model: JSSI-100C, JS Research Korea) at 30 °C for 72 h at 150 rpm. Then, the supernatant from the fermentation mixture was recovered by centrifugation at 8000× *g* for 10 min. The ethanol concentration in the supernatant of yeast culture was analyzed using gas chromatography-mass spectrometry (GC-MS) (Shimadzu, Tokyo, Japan) [19].

GC-MS analysis for ethanol quantification was carried out in Shimadzu GC-MS (Shimadzu, Tokyo, Japan) fitted with DB-wax column (Agilent J & W GC column, CA, USA). The analysis used helium as carrier gas with a column flow rate of 1.22 mL/min. The GC inlet was set in a split ratio of 30.0, and the temperature of the injector was maintained at 230 °C. The column oven temperature was controlled in a program with the initial holding temperature at 50 °C for 1 min. The temperature was then increased to 200 °C for a hold time of 5 min, at a ramping rate of 20 °C/min. The MS program was fixed with an ion source temperature of 200 °C and mass range from *m*/*z* 30 to 600 [34,35]. The ethanol concentration after fermentation was estimated from the obtained peak area compared to diluted absolute ethanol (99.8% *v*/*v*). The analysis was repeated three times.

### 2.7. Analysis of Chemical Changes in Biomass

Chemical compositions and chemical structural alterations of untreated and pretreated biomass were examined with an FTIR spectrometer (Spectrum 2000, Perkin Elmer, Waltham, MA, USA) to evaluate the impact of pretreatment on biomass. The analysis was conducted at a resolution ranging from 400 cm^−1^ to 1700 cm^−1^, and the spectral data were analyzed using Spectrum 2.00 software (Perkin Elmer, Waltham, MA, USA).

### 2.8. Analysis of Morphological Changes

The morphological and structural changes in the biomass before and after pretreatment was visualized through scanning electron microscopy (Model: JSM—5410LV, Jeol, Tokyo, Japan) at an accelerating voltage of 20 kV. Untreated and pretreated samples were prepared by attaching them to a specimen stub and coating them with gold before the inspection. The scanning electron microscope (SEM) images were visualized at a magnification of 100 μm and compared with the untreated sample.

## 3. Results and Discussion

### 3.1. EMIMAc Pretreatment and Its Optimization

Pretreatment is considered an essential step in a biorefinery in reducing the recalcitrance of the biomass. In the current study, pretreatment was performed using EMIM-Ac IL. Several studies carried out previously reported pretreatment with EMIM-Ac as efficient in various biomass and reported for more than 90% of sugar yield recovery from biomass [36,37,38,39]. Moreover, EMIM-Ac has the capability of rearranging hydrogen bonds efficiently in biomass, thus leading to cellulose dissolution [40]. Additionally, this IL is reported to drastically increase the saccharification rate as well. However, pretreatment, if not carried out under optimized conditions, may not increase the sugar yield significantly, emphasizing the importance of optimization.

Considering the necessity of optimized pretreatment condition, response surface methodology (RSM) was conducted to optimize pretreatment conditions for rice straw using IL. BBD was chosen to optimize the pretreatment conditions as it requires fewer runs than that of factorial design [41]. The design considered three pretreatment factors, including loading ratio of rice straw to EMIM-Ac (X_1_), pretreatment temperature (X_2_), and pretreatment time (X_3_) for optimization depending on the reducing sugar yield after pretreatment. These three pretreatment parameters were selected in this optimization experiment, as they are proved to be an important factor to determine the pretreatment efficiency, and they are easily adjusted during pretreatment operation [2]. The model suggested 17 runs, and the details of each run and its sugar yield are depicted in Table 2. It was noted that the smallest and largest reducing sugars obtained from the RSM experiment were 15.34 (Run No. 11) and 54.64 mg (Run No. 3), respectively, which was equivalent to a 3.56 fold-difference. This observation suggested the significance of optimization and efficiency of the RSM method.

The significance of the mathematical model was also statistically analyzed with ANOVA (Table 3). The model significance was confirmed with a *p*-value less than 0.05. Additionally, Table 3 show temperature as another significant term in this pretreatment. A similar optimization study conducted on rice straw pretreatment with IL also reports temperature as a significant factor in pretreatment [42]. Besides this, the model could also show the effect of interaction between factors on sugar yield (Figure 2). The sugar yield in the 3D plot increases with an increase in temperature and time.

In addition to this, the model was also able to predict the ideal pretreatment conditions for the highest sugar yield based on data from the experimental design (Table 3). Under the optimum pretreatment condition, the model predicted a sugar yield of 51.96 mg. Other experiments in the study followed the optimum pretreatment conditions predicted by the model.

### 3.2. Analysis of Morphological Changes

Morphological changes in the untreated and pretreated biomass were analyzed using SEM (Figure 3). This analysis facilitates observing any structural changes to the biomass after pretreatment. Hence, to confirm the effect of pretreatment, the biomass was subjected to microscopic analysis to elucidate the physical changes. The untreated biomass had a regular and tough surface. It was more fibrous in appearance and had a smooth surface. The morphology of untreated rice straw was more intact in comparison with the pretreated rice straw. Unlike the untreated biomass, pretreated rice straw appeared to be more disorganized and became more porous. These structural changes enhance the subsequent enzymatic hydrolysis. EMIM-AC was previously reported having the ability to boost the specific surface area of the lignocellulosic biomass by removing lignin and simultaneously reducing its crystallinity, ensuring an enhanced saccharification rate [37]. In the present study, pretreatment with IL generated structural changes in biomass, enhancing the specific surface area for enzymatic accessibility [36]. This porous biomass is more accessible to enzymes and thus can increase the sugar yield [38].

### 3.3. Effect of Different Anti Solvents and Recycled EMIM-Ac on Reducing Sugar Yields

Anti-solvents are essential in the post pretreatment process. They help to separate IL, in addition to the soluble lignin from pretreated solids. Several anti-solvents such as water, methanol, ethanol, isopropanol, etc., are studied to investigate their performance in delignification and saccharification yield. A study using IL pretreatment on wheat straw investigated the role of methanol, ethanol, and water as anti-solvents to separate pretreated solids [43]. However, this investigation could not find any significant effect of the anti-solvent in increasing sugar yield. A more recent study on investigating different parameters in protic IL pretreatment for ethanol generation studied the effect of anti-solvents in boosting sugar yield [44]. The study used water, ethanol, isopropanol, and isoamyl alcohol as anti-solvents. Even though different anti-solvents did not show significance in sugar yield, they could show a significant difference in delignification. The obtained results infer those anti-solvents with a low number of carbons in the alkyl chain lead to more interaction between hydrophobic lignin and alcohols and thereby allow us to enhance delignification [44].

In the present study, the effect of recycled EMIM-Ac, along with the usage of different anti solvents and their combinations in the washing step, was analyzed. The recyclability of IL is an important consideration as it can help in reducing the production cost [45]. There were previous studies on recycling EMIM-Ac. The effectiveness of recycled EMIM-Ac and different anti-solvents were evaluated based on saccharification yield (Figure 4). Anti-solvents, namely, water, acetone, methanol, and their combinations in 1:1 and 1:5 ratio (A1M5—Acetone:Methanol (1:5), M1A5—Methanol:Acetone (1:5)) were used in this study. Recycling studies were carried out for up to five recycles. The results showed that, in the initial pretreatment (R0), the saccharification yield was high when washing was carried out with solvents other than water. In the subsequent pretreatments carried out with recycled IL, the saccharification yield decreased with an increase in recycling number (R1 to R5) compared with R0. This is in line with the previous study, where recycled EMIM-Ac showed a decrease in sugar conversion after each recycle [26]. Sugar conversion decreased to <5 wt% between the 1st and 5th recycle, and it further decreased to 10–50 wt% between the 5th to 10th recycle. This reduction in sugar conversion was related to the accumulated lignin content in the solvent after each recycle and also to the presence of water in the solvent as the study used water in the washing process [26]. However, in this study, the sugar yield from the untreated rice straw was only 8.30 ± 1.061 mg/100 mg biomass, which clearly indicates the effectiveness of pretreatment even after five recycles.

The use of a suitable anti-solvent can also help in increasing the sugar yield. A previous study using water and water:acetone (1:1, *v*/*v*) as antisolvent for EMIM-Ac pretreated Miscanthus showed water:acetone (1:1, *v*/*v*) as a suitable anti-solvent in increasing sugar yield [46]. However, the exact mechanism correlating the anti-solvent and increased sugar yield was not very clear, and it was assumed that the anti-solvent helped in washing away lignin and inhibitors in regenerated biomass, leading to enhanced sugar yield [47]. Among the different anti-solvents used in this study, the biomass washed by pure methanol yielded the maximum reducing sugar (Figure 4). It could maintain the sugar yield between 57.31 to 45.88 mg even after five recycles with no statistical difference (*p*-value less than 0.05). The statistical studies also showed that recycling is efficient even after five recycles when methanol was used as an anti-solvent. This could be attributed to the polarity of the solvent. The reported polarity of the solvent boosted its effectiveness to remove IL [44]. Methanol being more polar than other solvents, could facilitate in removing more IL and lignin solubilization after pretreatment. This could enhance the enzyme to access cellulose without any non-specific binding [44]. Moreover, methanol, possessing a lower boiling point, can be easily separated from pretreatment slurry by rotary evaporation. Hence, ethanol fermentation studies were only conducted on pretreated biomass washed with methanol as an anti-solvent.

### 3.4. Analysis of Chemical Changes in Biomass

The chemical conformational changes that could occur in biomass after IL pretreatment were analyzed by Fourier transform infrared spectroscopy (FT-IR) analysis (Figure 5). The FTIR spectrum of raw rice straw, which corresponds to carbohydrate and aromatic derivatives before and after pretreatment under the optimal condition (5% loading ratio at 128.4 °C for 71.83 min), were comparatively evaluated (Table 4). The FTIR analysis confirms changes in the chemical structures of the components in lignocellulosic biomass when comparing the intensities of peaks. The band appearing near 897 cm^−1^ matches with the typical peak of β-glucosidic bonds in cellulose, increasing its intensity and confirming its exposure and cellulose swelling post pretreatment [46]. The alteration in peak intensity near 1060 cm^−1^ of the pretreated and untreated biomass suggests that the IL treatment process enhanced the ratio of cellulose in the biomass [48]. The reduction in the intensity of peak at 1246 cm^−1^ is indicative of lignin removal in rice straw [10]. The intensity changes in 1321 and 1460 cm^−1^ (syringyl, guaiacyl, and methoxy groups in lignin) imply that the IL pretreatment caused a breakage in the links within and between the lignin–carbohydrate complex. The strong bands at approximately 1373 cm^−1^ were the characteristics of the cellulose biosorption peak that might be due to substituted aromatic components of lignin on the rice straw surface [38]. The decrease in intensity near 1430 cm^−1^ (associated with bending vibration of CH_2_ group) represents cellulose suggesting the IL pretreatment could reduce crystalline cellulose [46]. The peaks at 1510 and 1637 cm^−1^ represent the damages caused to lignin after EMIM-Ac pretreatment. In fact, these changes depicted in FTIR data discloses the ability of IL in degrading the lignin content in the biomass [46,49]. These changes could also be noted in rice straw samples pretreated with recycled IL, even after five recycles, confirming the potential of EMIM-Ac recyclability (Figure 6).

### 3.5. Effect of Recycled EMIM-AC on Sugar Yield and Fermentation

The efficacy of the IL pretreatment with recycled EMIM-AC was assessed based on the enzymatic hydrolysis and bioethanol fermentation after pretreatment. Fermentation studies were conducted on both pretreated and untreated rice straws that use recycled EMIM-Ac and pure methanol as an anti-solvent. *Saccharomyces cerevisiae* TISTR 5606 was used for the fermentation of biomass. Figure 7 represent the ethanol production and fermentable sugar yield from the pretreated biomass after five recycles. The untreated rice straw could produce only 0.46 ± 0.07% (*v*/*v*) ethanol. However, after pretreatment, the ethanol production was increased to 0.75 ± 0.02%, which increased after using recycled IL for pretreatment. The ethanol production was 0.87 ± 0.03% even after the fifth recycle. This implies that the biomass pretreated with recycled IL was able to produce almost 1.9-fold more ethanol than the untreated biomass even after five recycles. This could be due to the presence of some residual anti-solvent in recycled IL, which facilitated more lignin removal [45]. Table 5 show a comparison of the present study with some previously reported ethanol yields from various biomass using EMIM-Ac. In this study, pretreated rice straw at R0 and R5 produced 63% and 89% higher ethanol, respectively, compared to untreated rice straw. The results clearly demonstrate the efficiency of recycled IL in pretreating the biomass and the importance of suitable anti-solvent. It also indicates the potential of IL to be used in industries for reducing the cost of production. However, further studies are required to confirm this effect when upscaling for industrial purposes.

## 4. Conclusions

The current study emphasized the requirement of an appropriate anti-solvent and recyclability of ionic liquid in pretreating rice straw for an objective of cost reduction. The optimum pretreatment conditions for rice straw were obtained using RSM. Besides, RSM predicted a pretreatment with a solid loading of 5% wt at a temperature of 128.4 °C for about 71.83 min as the optimum pretreatment condition for rice straw. Despite this, the study also identified pure methanol as an appropriate anti-solvent that can not only enhance lignin removal but also increase sugar yield. In addition to this, methanol, when used as an antisolvent, can be easily recycled and reused for further washing processes. Moreover, it provides an added advantage of reducing wastewater generation in biorefineries. The recycling studies of ionic liquid revealed that a higher ethanol yield (1.9 fold more than untreated biomass) could be achieved even after five recycles of ionic liquid. Altogether, this work demonstrates the potential of recycled ionic liquid and the importance of an appropriate anti-solvent in pretreating biomass, particularly pretreatment cost reduction. Additionally, the recyclability of ionic liquid could consequently cause a reduction in production costs for industrial-scale applications.

## Figures and Tables

**Figure 1 bioengineering-09-00115-f001:**
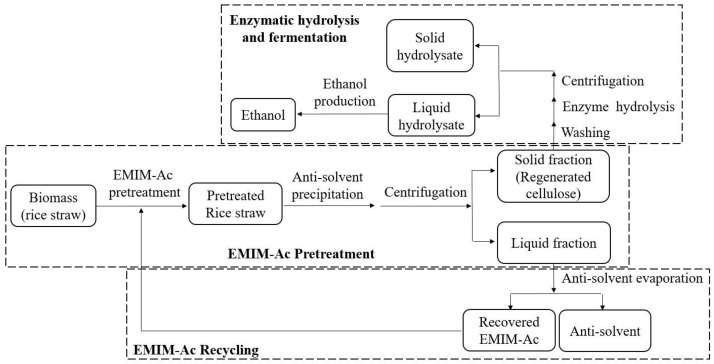
Process flow of EMIM-Ac pretreatment and enzymatic saccharification of biomass and process of EMIM-Ac recycling.

**Figure 2 bioengineering-09-00115-f002:**
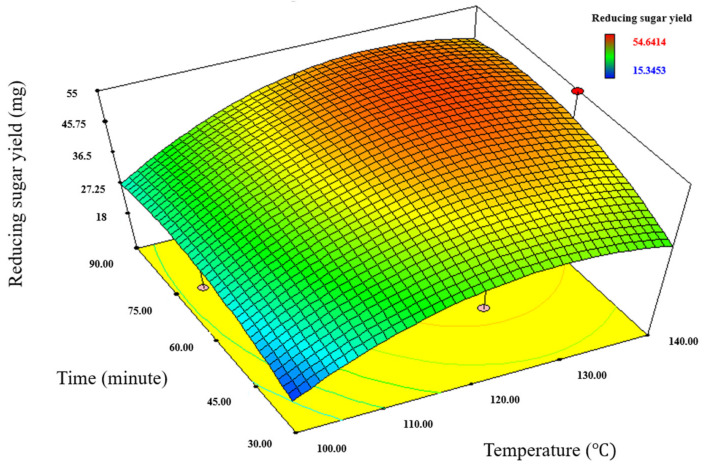
Contour plots representing the relation between the pretreatment factors on sugar yields using EMIM-Ac.

**Figure 3 bioengineering-09-00115-f003:**
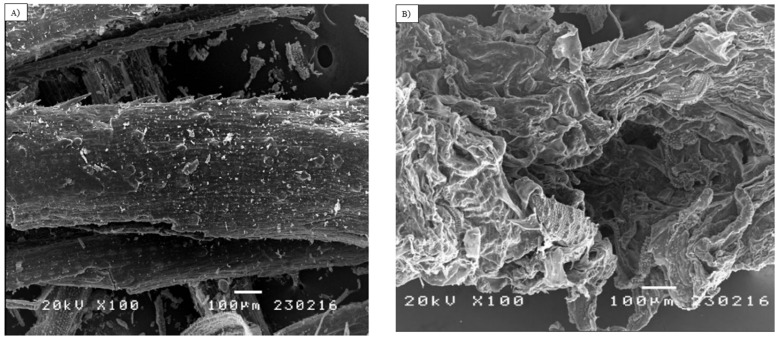
SEM images of biomass portraying the effects of EMIM-Ac pretreatment on biomass morphology. (**A**) Untreated rice straw and (**B**) EMIM-Ac pretreated rice straw using water as an anti-solvent at the optimal condition.

**Figure 4 bioengineering-09-00115-f004:**
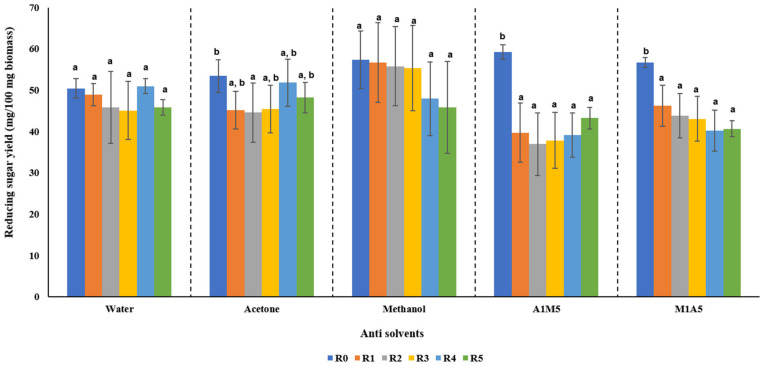
Reducing sugar yield attained from enzymatic saccharification of pretreated biomass with fresh and recycled EMIM-Ac and using various types of anti-solvent (A1M5—Acetone:Methanol (1:5), M1A5—Methanol:Acetone (1:5)). Alphabets indicate the results of ANOVA analysis, and different alphabet means significantly different (*p* < 0.05).

**Figure 5 bioengineering-09-00115-f005:**
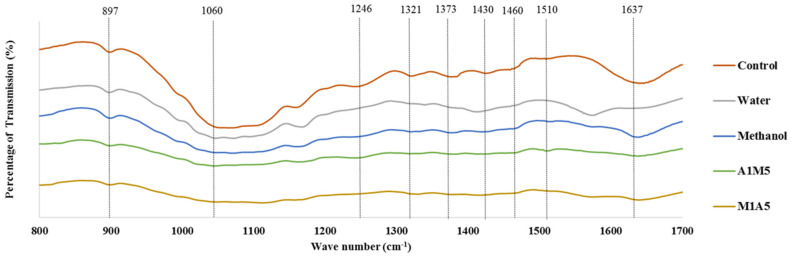
FT-IR spectra of untreated and pretreated rice straw for wavenumbers between 400–1700 cm^−1^ using EMIM-Ac pretreatment with various types of anti-solvents.

**Figure 6 bioengineering-09-00115-f006:**
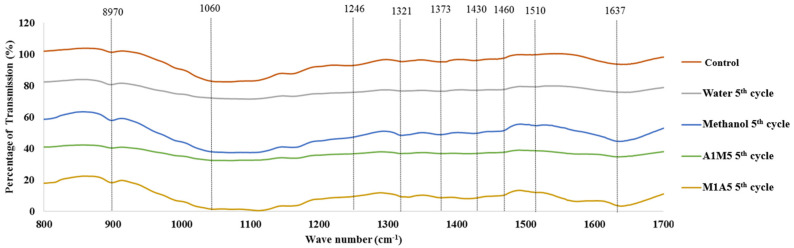
FT-IR spectra of untreated and pretreated rice straw for wavenumbers between 400–1700 cm^−1^ using EMIM-Ac pretreatment after five times recycle with various types of anti-solvents.

**Figure 7 bioengineering-09-00115-f007:**
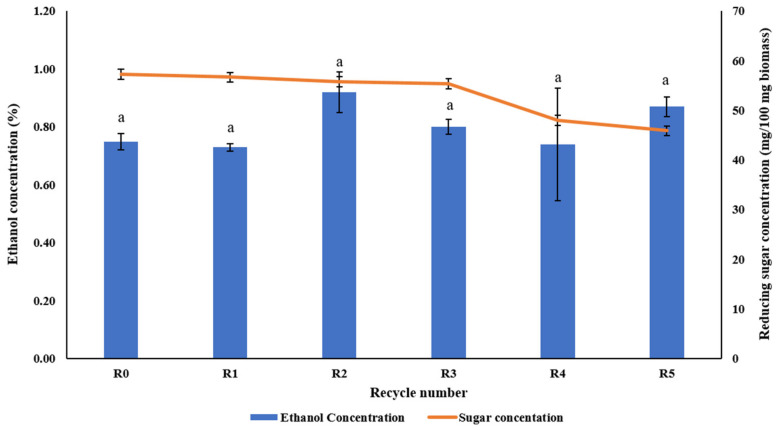
Ethanol concentration and reducing sugar yield obtained from enzymatic hydrolysis of pretreated rice straw by recycled EMIM-Ac and utilizing methanol as an antisolvent. Small alphabet indicate the results of ANOVA analysis, and different alphabet means significantly different (*p* < 0.05).

**Table 1 bioengineering-09-00115-t001:** Value of pretreatment parameters with corresponding coded level.

Pretreatment Factor	Level of Factor		
	Low	Med	High
Abbreviation	−1	0	1
Loading ratio (wt%) (X_1_)	5	10	15
Temperature (°C) (X_2_)	100	120	140
Time (min) (X_3_)	30	60	90

**Table 2 bioengineering-09-00115-t002:** BBD design to assess the effects of pretreatment factors (loading ratio (X_1_, %), temperature (X_2_, °C) and time (X_3_, min)) on reducing sugar yield (Y, mg) of EMIM-Ac pretreated rice straw.

Run	Pretreatment Condition			Reducing Sugar (Y) (mg)
	Loading Ratio (X_1_) (%)	Temperature (X_2_) (°C)	Time (X_3_) (min)	
1	15	120	30	22.36
2	5	120	90	44.19
3	5	140	60	54.64
4	10	120	60	48.31
5	15	120	90	45.14
6	10	120	60	43.52
7	10	120	60	45.70
8	15	100	60	16.99
9	10	120	60	46.57
10	5	120	30	32.48
11	10	100	30	15.35
12	15	140	60	29.95
13	5	100	60	23.09
14	10	120	60	44.10
15	10	140	90	31.90
16	10	140	30	40.70
17	10	100	90	27.98

**Table 3 bioengineering-09-00115-t003:** Optimum pretreatment conditions and predicted sugar yield obtained from RSM experiment.

EMIM-Ac Pretreatment	Mathematical models	Sugar content (mg) = - 429.80831 − (0.99874 × Conc.) + (7.03793 × Temp.) + (0.97107 × Time) − (0.027403 × Temp^2^) − (0.00676158 × Time^2^)
	Optimal pretreatment condition	5% loading ratio, 128.4 °C temperature, 71.83 min time
	Predicted sugar yield	51.96 mg

**Table 4 bioengineering-09-00115-t004:** The FTIR spectrum with different wavenumbers representing different functional groups in chemical derivatives of lignocellulosic biomass.

Peak, cm^–1^	Functional Group Assignment	References
897	β-glycosidic linkage; vibration of amorphous cellulose	[48,50]
1060	Bond Stretching in C–O of homo and heteropolysaccharide	[38]
1246	C–O stretching of phenolics in lignin	[46]
1321	Stretching vibration of C=O in syringyl, guaiacyl group	[33]
1373	Deformation of C–H in homo and heteropolysaccharide	[38]
1430	C–H_2_ bending of cellulose	[51]
1460	Deformations in C–H bonds of lignin	[52]
1510	Vibration in aromatic skeleton of lignin	[48]
1637	Phenolics in lignin	[46]

**Table 5 bioengineering-09-00115-t005:** Comparison with previous studies to produce ethanol from IL-pretreated lignocellulosic biomass with various types of ionic liquids and anti-solvents.

Biomass	Pretreatment Conditions	Anti-Solvent	Sugar Concentration	Ethanol Concentration	References
Wood powder	15% solid loading, 120 °C, 2 h	Dimethyl formamide	Glucose: 31 g/100 g biomass Xylose: 314.4 g/100 g biomass	3 g/L	[53]
Yellow pine wood	5% solid loading, 140 °C, 45 min	-	26.89 g/100 g biomass	2.6 g/L	[54]
Barely straw	5.26% solid loading, 105 °C, 7.5 h	Water	36.3 g glucose/100 g biomass 13.2 Xylose/100 g biomass	18.5 g/L	[55]
Water hyacinth	5.89% solid loading, 120 °C, 180 min	Water	4.5 g/100 g biomass	1.3 g/L	[56]
Rice straw	15% solid loading, 120 °C, 5 h	Water	44.3 g glucose/100 g biomass	1.92 g/L	[51]
Rice straw	R0, 5% solid loading, 128.4 °C, 71.83 min	Methanol	57.3 mg/100 mg biomass	5.9 g/L	This study

## Data Availability

The authors confirm that the data supporting the findings of this study are available within the article.

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
