# Peer review of "Improvement of Enzymatic Saccharification and Ethanol Production from Rice Straw Using Recycled Ionic Liquid: The Effect of Anti-Solvent Mixture"

_bioengineering, 2022, doi:10.3390/bioengineering9030115_

Round 1
Reviewer 1 Report
The article "Improvement of enzymatic saccharification and ethanol production by recycled ionic liquid-solvent mixture pretreatment of rice straw" is very interesting and written well. For this research, they used recycled ionic liquid (EMIM-Ac) for the pretreatment of rice straw biomass and also predicted a model. In addition they also evaluated the anti-solvent and found that methanol is the most suitable for enhancing the sugar yield.
The article introduction was written well by giving a nice overview about the topic. Also the methods were described well. The results were presented well by visualizing the data. A nice discussion was provided by comparing the results of similar studies. However, there are so many unnecessary self citations (18 self citations - Sriariyanun, M; 7 self citations - Chuetor, S and more than 5 self citations from some of the other authors ) were found in the article. According to ethical publication guidelines, the percentage of self-citations should not exceed 10-12%. I strongly recommend to remove the unnecessary self citations. Hence I recommend to publish it after removing the self citations.
Minor points:
Figure 4: What are A1M5 and M1A5, please mention clearly in the legend.
Figure 4 is showing no significance difference between the water and ethanol as the antisolvent, but in the abstract it was mentioned that methanol is the best suitable solvent. why?
Author Response
General comment: The article "Improvement of enzymatic saccharification and ethanol production by recycled ionic liquid-solvent mixture pretreatment of rice straw" is very interesting and written well. For this research, they used recycled ionic liquid (EMIM-Ac) for the pretreatment of rice straw biomass and also predicted a model. In addition, they also evaluated the anti-solvent and found that methanol is the most suitable for enhancing the sugar yield.
The article introduction was written well by giving a nice overview about the topic. Also the methods were described well. The results were presented well by visualizing the data. A nice discussion was provided by comparing the results of similar studies. However, there are so many unnecessary self-citations (18 self-citations - Sriariyanun, M; 7 self-citations - Chuetor, S and more than 5 self-citations from some of the other authors) were found in the article. According to ethical publication guidelines, the percentage of self-citations should not exceed 10-12%. I strongly recommend to remove the unnecessary self-citations. Hence I recommend to publish it after removing the self-citations.
Author’s response: Thank you very much for your valuable comment. The unnecessary self-citations have been removed.
Comment 1: Figure 4: What are A1M5 and M1A5, please mention clearly in the legend.
Author’s response: Thank you very much for your valuable comment. The figure legend added in Line no: 350-351.
Comment 2: Figure 4 is showing no significance difference between the water and ethanol as the anti-solvent, but in the abstract it was mentioned that methanol is the best suitable solvent. why?
Author’s response: Thank you very much for your valuable comment. The best anti solvent was chosen based on maximum sugar yield produced when washing with anti-solvents. The sugar yield obtained using water as an anti-solvent was 50.434 mg/100 mg biomass, whereas washing with pure methanol yield 57.313 mg reducing sugar per 100 mg biomass. Also, after each recycle, the sugar yield achieved from washing with methanol was higher than when washing with water. In addition to this, methanol is easy to recover by rotary evaporation at around 40 ℃. However, recovery of water require higher temperature as its boiling point is 100 ℃, consequently, it needs a high energy consumption for anti-solvent recovery process. Hence, methanol was chosen as the best anti-solvent in the study. The sugar yield attained using water and methanol as anti- solvent is given in the table below for your reference.
Sugar yield using water and methanol as anti-solvent
|
Recycle number |
Reducing sugar yield ( mg/100 mg biomass) |
|
|
Water |
Methanol |
|
|
R0 |
50.434 ± 2.3 |
57.313 ± 6.9 |
|
R1 |
48.939 ± 2.6 |
56.684 ± 9.5 |
|
R2 |
45.841 ± 8.7 |
55.814 ± 9.5 |
|
R3 |
45.103 ± 7.0 |
55.328 ± 10.3 |
|
R4 |
50.996 ± 1.8 |
47.971 ± 8.8 |
|
R5 |
45.901 ± 1.8 |
45.877 ± 11.1 |
Reviewer 2 Report
The present manuscript is intended to optimize saccharification and ethanol production from IL-pretreated rice straw. In general, the main purpose of the study is not well motivated when considering the related literature. The result and discussion section is mainly descriptive, focusing on describing the results obtained and therefore lack on a critical discussion of the corresponding results with related references. In addition, this work has been designed without considering common strategies for ethanol production (such as working at high substrate concentrations for increasing final ethanol titers), thus limiting the impact of the study. The following comments should be considered prior to its publication:
- Line 57: a brief description about the main chemical inhibitors and how they are produced is needed to fully understand this statement.
- Line 191: Phosphate buffer 4.7 was used for enzymatic hydrolysis. However, the buffer capacity of phosphate buffer ranges between pH 5.8 and pH 8.0. Authors should better explain why this buffer was used, instead of acetate buffer or citrate buffer, which are commonly used for enzymatic hydrolysis of lignocellulose.
- A major critical point of this study is that enzymatic hydrolysis processes have been performed at substrate loadings as low as 2.5% w/v. This is far from the substrates concentrations requires to reach a cost-competitive ethanol production processes (substrate concentrations above 20% w/w). Indeed, fermentation assays were supplemented with 1% glucose in order to evaluate ethanol production. Authors should critically describe why such a low substrate concentration was used and discussed what are the major challenges to work with IL-pretreated biomass at higher substrate concentrations.
- Line 209: what was the procedure to remove sodium azide?
- Line 331: abreviations for A1M5 and M1A5 must be define both in the main text and in figure legend 4.
- Line 343: should not be Figure 4 instead of Figure 3?
- Figure 4: in addition to show statistic between data from recycling cycles, it is important to assess statistics in R0 among all tested antisolvents to support the concluding statements.
- Ethanol production ranged between 0.46% and 0.87% (v/v). Authors should critically discussed about the scientific impact of this study rather than compared numbers between them. Indeed, 1% glucose was added prior to fermentation. Such supplementation hinders the comparison between assays in terms of ethanol production, as it is difficult to evaluate ethanol yields from hydrolysate sugars.
- Table 5: columns listing sugar and ethanol yields mainly include sugar and ethanol concentrations instead. This must be double checked in order to include data accordingly.
Author Response
General comment: The present manuscript is intended to optimize saccharification and ethanol production from IL-pretreated rice straw. In general, the main purpose of the study is not well motivated when considering the related literature. The result and discussion section is mainly descriptive, focusing on describing the results obtained and therefore lack on a critical discussion of the corresponding results with related references. In addition, this work has been designed without considering common strategies for ethanol production (such as working at high substrate concentrations for increasing final ethanol titers), thus limiting the impact of the study. The following comments should be considered prior to its publication:
Author’s response: Thank you very much for your valuable comments and suggestions. The authors would like to clarify that the intention of this study is not limited to optimizing saccharification and ethanol production, but also to determine the recyclability of IL and thereby reduce the cost of pretreatment in its industrial application. The study could also confirm the importance of anti-solvent in the washing process. Few studies have been added in the Section: 3.3, Line no: 331-332, 340- 345, 352-359.
Also, this study was conducted in a experimental scale and the scaling up of this work will be done in future for analyzing its industrial implementation and economic analysis.
Comment 1: Line 57: A brief description about the main chemical inhibitors and how they are produced is needed to fully understand this statement.
Author’s response: Thank you very much for your valuable comment. A brief description about inhibitors is added in the manuscript. (Section: Introduction, Line no: 59-60).
Comment 2: Line 191: Phosphate buffer 4.7 was used for enzymatic hydrolysis. However, the buffer capacity of phosphate buffer ranges between pH 5.8 and pH 8.0. Authors should better explain why this buffer was used, instead of acetate buffer or citrate buffer, which are commonly used for enzymatic hydrolysis of lignocellulose.
A major critical point of this study is that enzymatic hydrolysis processes have been performed at substrate loadings as low as 2.5% w/v. This is far from the substrates concentrations requires to reach a cost-competitive ethanol production processes (substrate concentrations above 20% w/w). Indeed, fermentation assays were supplemented with 1% glucose in order to evaluate ethanol production. Authors should critically describe why such a low substrate concentration was used and discussed what are the major challenges to work with IL-pretreated biomass at higher substrate concentrations.
Author’s response: Thank you very much for your valuable comment. The authors would like to clarify that the phosphate buffer was written by mistake and we confirm that the enzymatic saccharification study has used citrate buffer in it. The mistake is corrected in the manuscript. (Section 2.5, Line no: 194).
This research work was conducted as an experimental study in the lab for analyzing recyclability and efficiency of recycled IL in ethanol production and hence used low substrate loadings. In spite of this, there are several other research works also in which they have used the same substrate loading (2.5% w/v) for enzymatic saccharification. Kindly check the links below for your reference.
https://pubs.rsc.org/en/content/articlelanding/2013/gc/c3gc40352a
https://www.sciencedirect.com/science/article/pii/S0960852413006950
https://www.mdpi.com/2073-4344/11/11/1274/htm
https://www.sciencedirect.com/science/article/pii/S0926669016302990
https://www.sciencedirect.com/science/article/pii/S2589014X22000433
However, we consider your reasonable opinion and will include high substrate loading in our further upscaling study. We agreed with you that high loading biomass is very important strategies to reach high concentration of sugars as reported by this publication previously https://biotechnologyforbiofuels.biomedcentral.com/articles/10.1186/1754-6834-6-52.
Also, the authors would like to mention that 1% glucose was supplemented into fermentation media to help yeast for its initial metabolism and acclimatization. A control with 1% glucose was used in the experiment and the ethanol produced by this control was reduced from the other fermentation samples. We also follow the sugar supplements as done in other related publications. Kindly check the links below for your reference.
https://link.springer.com/article/10.1007/s00449-017-1881-0
https://www.sciencedirect.com/science/article/abs/pii/S0019452221002648?via%3Dihub
https://www.mdpi.com/2306-5354/8/11/171/htm
https://www.sciencedirect.com/science/article/pii/S0960852421009378?via%3Dihub
http://ojs.kmutnb.ac.th/index.php/ijst/article/view/5238
Comment 3: Line 209: What was the procedure to remove sodium azide?
Author’s response: Thank you very much for your valuable comment. No specific procedure was adopted to remove sodium azide from hydrolysis mixture. Sodium azide was added into hydrolysis mixture to avoid any microbial contamination during the digestion. However, to prepare hydrolysate for ethanol fermentation, sodium azide was not added during saccharification.
Comment 4: Line 331: Abbreviations for A1M5 and M1A5 must be define both in the main text and in figure legend 4.
Author’s response: Thank you very much for your valuable comment. Abbreviations has been added in the manuscript. (Section: 3.3, Line no: 335 and Line no: 350-351).
Comment 5: Line 343: should not be Figure 4 instead of Figure 3?
Author’s response: Thank you very much for your valuable comment. The figure no has been corrected in the manuscript. (Section 3.3, Line no: 359).
Comment 6: Figure 4: In addition to show statistic between data from recycling cycles, it is important to assess statistics in R0 among all tested anti-solvents to support the concluding statements.
Ethanol production ranged between 0.46% and 0.87% (v/v). Authors should critically discussed about the scientific impact of this study rather than compared numbers between them. Indeed, 1% glucose was added prior to fermentation. Such supplementation hinders the comparison between assays in terms of ethanol production, as it is difficult to evaluate ethanol yields from hydrolysate sugars.
Author’s response: Thank you very much for your valuable comment. The statistics in R0 among all anti-solvents was assessed using IBM SPSS v2016 software and it shows that the sugar yield obtained using acetone, methanol and M1A5 as insignificant different. However, when comparing the sugar yield obtained after recycles using different anti-solvent, shows more sugar yield when washed with methanol. Hence methanol was concluded as suitable anti-solvent.
The impact of this study has been included in the manuscript. (Section 3.5, Line no: 424-427).
The authors would like to mention that 1% glucose was supplemented into fermentation media to help yeast for its initial metabolism and acclimatization in the hydrolysate. A control with 1% glucose was used in the experiment and the ethanol produced by this control was reduced from the other fermentation samples.
Comment 7: Table 5: Columns listing sugar and ethanol yields mainly include sugar and ethanol concentrations instead. This must be double checked in order to include data accordingly.
Author’s response: Thank you very much for your valuable comment. The column names have been changed in the manuscript. Also, the values have been converted to same units. (Section: 3.5, Line no: 432)
Reviewer 3 Report
The research articles lacks the discussion section because the recycling of ionic liquid from the biofuel production has been intensively studied and some examples could be integrated in the present article.
The title should be modified and should sound more specific.
Why do you use ionic liquid? Would it be possible to prepare a cost estimate if you use another type of feedstock treatment? Would it possible to perform the life cycle analysis using your data?
The conclusion is not very specific and does not underline the novelty of the present research.
Would it be possible to comment on the commercial value of the project? Is it possible to upscale or it will be always on a lab scale?
Author Response
General comment: The research articles lacks the discussion section because the recycling of ionic liquid from the biofuel production has been intensively studied and some examples could be integrated in the present article.
Author’s response: Thank you very much for your valuable comment. Few studies have been included in the Section: 3.3, Line no: 331-332, 340- 345, 352-359.
Comment 1: The title should be modified and should sound more specific.
Author’s response: Thank you very much for your valuable comment. We modified our title to be “Improvement of enzymatic saccharification and ethanol production from rice straw by recycled ionic liquid: Effect of anti-solvent mixture”
Comment 2: Why do you use ionic liquid? Would it be possible to prepare a cost estimate if you use another type of feedstock treatment? Would it possible to perform the life cycle analysis using your data?
Author’s response: Thank you very much for your valuable comment. The authors would like to clarify that ionic liquid was chosen in the study due to its properties of low volatility and recyclability since recycling was one of the objective in this study. Moreover, there have been researches suggesting ionic liquid as efficient in pretreating biomass. In this study, we have focused only on recycling IL and bioethanol production. We have not carried out any techno-economic assessment and life cycle analysis in this study. However, for our further scaling-up process, we will include the techno-economic assessment and environmental assessment per your kind suggestion and we agreed that this could be one important step to develop the feasible process.
Comment 3: The conclusion is not very specific and does not underline the novelty of the present research.
Author’s response: Thank you very much for your valuable comment. The conclusion has been modified. (Section 5, Line no: 441-445).
Comment 4: Would it be possible to comment on the commercial value of the project? Is it possible to upscale or it will be always on a lab scale?
Author’s response: Thank you very much for your valuable comment. Even though the lignocellulosic biomass has the potential to produce a wide variety of value-added products or chemicals, in this study we have focused on only bioethanol production. Most of the studies also currently focus on producing bioethanol from biomass in the biofuel community. However, there are studies coming up in pilot scale trying to produce various other value added chemicals like itaconic acid, lactic acid, vanillin etc. with much value than bioethanol. Though, they are just in its infancy stage and not yet to be in industrial scale.
There is another attempt to reach commercial value of this IL-mediated process, which is the high biomass loading, please see https://biotechnologyforbiofuels.biomedcentral.com/articles/10.1186/1754-6834-6-52. This group of research tested up to 50% biomass loading, with their aim to reach high concentration of sugar product.
In fact, this is a preliminary study trying to set up a biorefinery platform for lignocellulosic biomass valorization. And of course, there are several 2nd generation bioethanol plants worldwide in operation now, although with different pretreatment method. We totally agreed with reviewer suggestions that techno-economical analysis should be conducted for scaling up work to further reach the application in industrial scale. Currently, we have just also started next level of techno-economic and environmental assessment studies to upscale the production production of biofuels and other high value-added products from lignocellulosic biomass with IL-mediated process.
Round 2
Reviewer 1 Report
The article "Improvement of enzymatic saccharification and ethanol production from rice straw by recycled ionic liquid: Effect of antisolvent mixture" is improved compared to the previous version and the authors have been implemented all my previous suggestions. Hence I recommend the article for publication in the present form.
Reviewer 2 Report
No comments
Reviewer 3 Report
all comments were implemented